# Reducing 3D Hydrogel Stiffness, Addition of Oestradiol in a Physiological Concentration and Increasing FSH Concentration Improve In Vitro Growth of Murine Preantral Follicles

**DOI:** 10.3390/ijms241512499

**Published:** 2023-08-06

**Authors:** Mengxue Zheng, Jesús Cadenas, Susanne Elisabeth Pors, Tasnim Esa, Stine Gry Kristensen, Linn Salto Mamsen, Cristina Subiran Adrados, Claus Yding Andersen

**Affiliations:** 1Laboratory of Reproductive Biology, The Juliane Marie Centre for Women, Children and Reproduction, Copenhagen University Hospital, 2100 Copenhagen, Denmark; mengxue.zheng@regionh.dk (M.Z.); jesus.cadenas.moreno@regionh.dk (J.C.); sup@gubra.dk (S.E.P.); stine.gry.kristensen@regionh.dk (S.G.K.); linn.salto.mamsen@regionh.dk (L.S.M.); cristina.subiran.adrados@regionh.dk (C.S.A.); 2The Department of Clinical Medicine, Faculty of Health and Medical Science, University of Copenhagen, 2200 Copenhagen, Denmark; 3The Department of Biotechnology and Biomedicine, Technical University of Denmark, 2800 Kongens Lyngby, Denmark; tasnimesa@icloud.com

**Keywords:** murine follicle culture, three-dimensional culture, alginate, oestradiol (E_2_), follicle-stimulating hormone (FSH)

## Abstract

This study aimed to optimise culture conditions for murine preantral follicles to improve their growth and survival. Preantral follicles (diameter 100–130 µm) were isolated from prepubertal NMRI mice and individually cultured within alginate beads for 12 days. Three conditions were evaluated: (1) follicle re-encapsulation on day 6 of culture-reducing alginate concentration (0.5% to 0.25% *w*/*v*), (2) the presence of oestradiol (E_2_), and (3) increased follicle-stimulating hormone (FSH) concentration in the culture medium (from 10 to 100 mIU/mL FSH). Follicle morphology and growth, as well as anti-Müllerian hormone (AMH) production, were evaluated. From day 8, re-embedded follicles had a larger average diameter compared to follicles without alginate re-encapsulation (0.5% and 0.25% groups, *p* < 0.05). Oestradiol (1 µM) had a significantly positive effect on the mean follicular diameter and antrum formation (*p* < 0.001). Moreover, follicles cultured with 100 mIU/mL FSH showed faster growth (*p* < 0.05) and significantly higher antrum formation (*p* < 0.05) compared to the low FSH group. Nevertheless, AMH production was not affected by any of the culture conditions. In conclusion, the growth and survival of mouse preantral follicles during a 12-day period were improved by altering the alginate concentration midways during culture and adding E_2_ and FSH to the culture medium.

## 1. Introduction

Follicle culture techniques have provided important insights into the mechanisms regulating follicle growth and development [1,2,3]. Suitable culture conditions are essential to delineate the effects of specific hormones, factors, and molecules that promote the best follicle growth during different developmental stages. Moreover, the development of a platform to culture ovarian follicles may, in the long run, rescue some follicles from the naturally occurring atresia, maximise the number of potentially fertilisable oocytes for in vitro embryo production in livestock, and have significant clinical implications for restoring or preserving fertility in women, especially when combined with ovarian tissue cryopreservation [4,5].

Maintaining follicle morphology intact during culture has been described to be fundamental for its survival [2,6,7]. Hence, it is probably necessary to use an appropriate scaffold that acts as a flexible sponge around the follicle, that on one side provides a three-dimensional (3D) support and on the other side allows expansion and growth of the follicle. The 3D scaffold should closely mimic the ovarian environment and is likely to provide a more physiological input to follicle development than 2D cultures [2,7]. Alginate, a widely used material in 3D hydrogel culture systems, can form a network structure similar to the extracellular matrix [2,7,8]. It is non-toxic and biocompatible, and its physical qualities are maintained after gelation. Alginate scaffolds have shown potential for promoting follicle development in mouse follicle culture [2,4,7] by providing support for the follicle to maintain its 3D spherical structure, allowing contact and communication between somatic cells and oocytes but also enabling the follicles to exchange oxygen, various required nutrients, and biologically active substances with the external medium. 

Several studies have shown that different alginate stiffness (i.e., alginate concentration) affects follicle survival, growth, and antral formation, as well as somatic cell proliferation and differentiation [9,10,11]; smaller earlier stage follicles require higher concentrations of alginate, while larger follicles prefer lower concentrations [5,7,12]. This is consistent with the stromal microenvironment during follicular growth, where small immature follicles reside in the stiff ovarian cortex, and activated follicles migrate toward the less rigid medulla [3,13]. However, it is not yet clear whether an exchange of alginate concentration during in vitro growth will allow for an enhanced growth rate [4,7,14]. Therefore, we evaluated whether exchanging and using a new and less stiff alginate scaffold halfway through the culture could provide conditions that augment follicle growth.

Oestradiol (E_2_) and follicle-stimulating hormone (FSH) are essential for follicular growth. Oestradiol, synthesised by granulosa cells, is a key intrafollicular factor that affects granulosa cell proliferation [15,16]. However, there are clear species differences in the effect of E_2_ as a granulosa cell mitogen [17]. Human follicles expand during gonadotropin stimulation without oestrogen, which contrasts with mice, where E_2_ has been shown to stimulate the growth of preantral and antral follicles in hypophysectomised animals [18].

FSH stimulates the growth and development of ovarian follicles. However, in rodents, FSH alone is not sufficient to promote complete follicle and granulosa cell differentiation [19,20]. Instead, the synergistic actions of E_2_ and FSH are required to increase follicular proliferation and differentiation as well as decrease follicular atresia [18,19,20,21].

Since only a minor fraction of preantral follicles survive to the preovulatory stage in vivo, our study focused on optimising culture conditions to support the growth and survival of murine preantral follicles, intending to improve our understanding of the requirements for early follicle growth and survival in vitro. To do so, we explored whether re-encapsulation by reducing the alginate concentration halfway through the culture period improved follicle growth and evaluated the effect of E_2_ and FSH addition to the mouse follicle culture medium. 

## 2. Results

### 2.1. Effect of Oestradiol and Follicle Re-Embedding Halfway through the Culture

A total of 633 follicles were cultured in Experiment I. Growth and development were recorded by taking photographs every other day (Figure 1A–C). The average follicle diameters were compared among treatments at each time point (Table 1, Figure 2). No significant differences were found in mean follicle diameter among groups during the first 6 days of culture (all *p* values > 0.05). In general, follicles embedded in 0.5% alginate and cultured in E_2_-free medium showed the poorest growth and antrum formation during the second half of culture, whereas those re-embedded in 0.25% alginate and cultured with E_2_ showed a significantly increased diameter on days 10 and 12, accompanied by the highest rate of antrum formation compared to the other treatments (Table 1, Figure 2). 

### 2.2. Addition of Oestradiol in a Physiological Concentration Promotes Follicle Growth

The average diameter of follicles cultured in E_2_-containing medium increased by 10, 19 and 23 µm (95% CI = [4; 16], [13; 25], [16; 30]) compared to groups without E_2_ on days 8, 10 and 12, respectively, (all the *p*-values < 0.001, Table 2). Moreover, follicles cultured with a similar concentration of alginate formed an antrum significant in E_2_-containing media except for 0.5% alginate groups (Table 1).

### 2.3. Reduction of Alginate Concentration during Culture Can Promote Follicle Growth In Vitro

The average diameters of follicles that were re-embedded in alginate (0.5–0.25%) were larger than those in which alginate was not changed (0.5% and 0.25% groups) from day 8 of culture (all the *p*-values < 0.05, Table 2). The difference in average follicle diameter between follicles which were re-embedded, and the 0.25% alginate groups was 16 µm and 22 µm (95% CI = (6; 26) and (11; 33); both *p*-values < 0.05) higher on days 10 and 12 of the culture period, respectively (Table 2). The mean follicle diameter of re-embedded follicles compared to 0.5% alginate groups was increased by 10 µm, 24 µm and 29 µm (95% CI = (2; 18), (15; 33) and (18;39); all the *p*-values < 0.05) on days 8, 10, and 12 (Table 2). Although it revealed that re-embedding was beneficial for follicle growth, the survival rates were lower than that of the 0.25% alginate groups (*p*-values < 0.05, Table 1).

### 2.4. Concentration of Anti-Müllerian Hormone (AMH) in the Condition Medium

Overall, the concentration of alginate with or without the addition of E_2_ had no statistically significant effects on AMH secretion (Table 3 and Table 4), but on days 10 and 12, the 0.25% alginate group had a significantly higher AMH level than the 0.5% alginate group, with *p*-value = 0.016 (95% CI = (0.1; 1.1), Table 4). The AMH concentration below the lower limit of detection was considered an invalid test. The overall detection rate for samples tested for AMH was 65%. Even though the differences in detection rates were considerable, there were no statistically significant differences between groups at each time point (Table 4).

### 2.5. Increased Level of FSH Promotes the Growth of Secondary Follicles in Culture

To investigate whether increasing FSH concentrations (from 10 to 100 mIU/mL) would improve follicular development, the experimental data from Experiment II were compared with those from the previous Experiment I group of alginate re-encapsulation, the addition of E_2_, and low concentrations of FSH.

An increase in FSH concentration from 10 mIU/mL to 100 mIU/mL led to a statistically significant rise in follicle growth, as well as an increase in the survival rate and antrum formation (Table 5). Follicles in the high FSH group expanded more rapidly during the whole culture period than those in the low FSH group (all *p*-values < 0.05, Table 5 and Figure 3), reaching a maximum size on the final day, which was 1.45 times larger than that in the low FSH group. The high FSH group also showed an overall better numerical survival rate than the low FSH group, although not significant (*p*-value = 0.56). Furthermore, antrum formation occurred more often in the high FSH group, taking place in 82% of follicles versus 20% in the low FSH group, which was statistically different (*p*-value = 0.00005).

## 3. Discussion

This study identifies three parameters that optimise 3D mouse preantral follicle cultures: the addition of E_2_ in a concentration of 1 µM to the culture medium, the renewal of the alginate scaffold reducing the concentration to half midway through the culture period, and the increase in FSH concentration from 10 to 100 mIU/mL throughout the culture period. 

Oestradiol contributes to the growth of mouse preantral follicles and acts as a mitogen in granulosa cells [16]. However, the effect of E_2_ was only visible when the concentration was high at 1 µM, while 10 nM did not result in increased follicular growth. This probably reflects that the 1 µM concentration resembles the physiological concentrations observed in follicular fluid [16] which may be required to induce granulosa cell mitosis and lead to growth promotion. On the other hand, it has also been described that mouse follicle development during 2D conditions does not require physiological E_2_ concentrations [22,23], even one study [24] suggested that a high concentration of 1 µM E_2_ induced abnormal follicular development. The environment of a 3D scaffold used in the present study may promote E_2_-receptor expression [19,25,26] and, in this way, E_2_-introduced conditions favour follicle development. Auto- and paracrine signalling induced by the E_2_ is likely to be facilitated by an intact spherical follicles structure and promote growth. Furthermore, E_2_ can synergistically cooperate with FSH to induce the formation of the antral cavity, as well as the expression of *cyp19*, *aromatase*, and *LH receptors* [18,27,28,29,30]. 

Among the three tested conditions for follicle embedding, replacing the 0.5% alginate scaffold with a lower concentration of alginate (0.25%) halfway through the culture resulted in significantly improved follicular growth compared to maintaining the concentration constant throughout the culture period. The growth advantage of follicles with alginate replacement became apparent already two days after re-embedding, on day 8 of culture. Thus, the follicular diameters were significantly increased compared to the 0.25% and 0.5% groups, implying the importance of a low-concentration alginate replacement rather than the actual alginate concentration. This suggests that the alginate became compressed during culture and even though it provided a 3D scaffold for the follicles, it also limited their growth, probably by a lack of flexibility, even in the 0.25% alginate group. While the alginate is accommodating, it lacks autonomous dissolution or sufficient deformability, thus limiting the available growth space for the follicles, which may result in follicle compression. This suggests that new 3D scaffolds should be more flexible or be able to be degraded by the follicles as they increase their diameter with continuous 3D supports.

Furthermore, despite the slightly larger follicle diameter of the 0.25% alginate group compared to the 0.5% alginate group on day 8, this advantage did not persist throughout the subsequent days of culture. In fact, there was no difference in follicular growth between the 0.5% and 0.25% alginate groups, which is consistent with the findings of Xu et al. [10], despite AMH secretion on days 10 and 12 in the 0.25% alginate group was higher than in the 0.5% alginate group.

In general, alginate re-encapsulation combined with 1 µM E_2_ best promoted the growth of murine secondary follicles. By the 12th day of culture, the average diameter of follicles in this group had increased by more than twice the initial size, significantly larger than the follicles cultured in the other five groups, which had a final mean diameter of up to 1.8 times that at the beginning (Table 1 and Figure 2A). Moreover, in contrast to the continuous growth of follicles re-embedded with E_2_ supplementation, the average follicle diameter in all other groups started to decrease from day 10, probably reflecting the start of atretic changes.

Although secondary follicles re-embedded and cultured with E_2_ performed best during 12 days of culture, they ended up with an average diameter of 233 μm, a 205% increase from the start. This finding is similar to that reported by Rajabi and co-workers [31], who found a 219% increase in the mean diameter of follicles from NMRI mice (with an initial diameter of 120–140 μm) encapsulated in 0.25% alginate and cultured for 12 days. However, Xu and co-workers [10] reported a final diameter of 326 μm (with an initial diameter of 100–130 μm) after 12 days of culture, which is larger than we found. This difference is most likely explained by differences in media composition: Xu and co-workers used 3 mg/mL bovine serum albumin (BSA) and 1 mg/mL fetuin during the culture, while we employed 5% foetal bovine serum (FBS). Evidence in mice and humans has demonstrated that various serum-derived products impact preantral follicle growth differently in vitro [32,33,34]. In particular, FBS contains more than a thousand components, including proteins, hormones, growth factors, enzymes, and other undefined constituents, which generate potential batch-to-batch variations, making comparing results from different studies difficult. Equally important, mouse strains used for research purposes exhibit varying sensitivities to various hormones, including gonadotropins [35], which may account for the diverse results of the different mouse strains.

In addition, the follicle survival rate in the re-encapsulated alginate group was lower than in the non-re-encapsulated groups, especially in the 0.25% alginate group. This may be attributed to the side effects of the alginate gel re-embedding process, which involves artificial manipulation, that may inadvertently harm the follicles, resulting in higher rates of subsequent death and degeneration during continued culture. In addition, the alginate lyase solution used to perform alginate re-encapsulation is low in nutrients and may potentially attenuate follicle survival due to the 30-min incubation period. Furthermore, despite our efforts to maintain a constant temperature, humidity, and CO_2_ concentration during the procedures, inevitably, the environmental conditions inside the incubator could not be perfectly replicated. 

The concentration of FSH affects follicular development. In our study, it was evident that the average diameter and antrum formation rate of secondary follicles cultured with 100 mIU/mL FSH were significantly better than those cultured with 10 mIU/mL FSH. This is consistent with the findings of Hardy and co-workers, who found improved growth of follicles cultured with a higher FSH concentration compared to those cultured at a low FSH concentration [36]. According to Nayudu and colleagues [37], a slow growth rate of follicles during the early stage of culture can negatively impact antral formation, which aligns with our observations where the average growth rate of follicles cultured with 10 mIU/mL FSH was approximately 9.5 μm/day from day 2 to day 4, while those cultured in high FSH medium exhibited a growth rate of 20.5 μm/day, the corresponding antrum formation rates were 20% and 80%, respectively.

Our results indicate that secondary follicles exhibit a rapid growth rate starting from the second day of culture in the presence of high concentrations of FSH, with a dramatic increase in follicle diameter between days 6 and 8 of culture. Probably, at that point, the granulosa cells had increased FSHR expression and became more sensitive to FSH, promoting the formation and expansion of the antral cavity. During the last days of culture, a follicle growth plateau was observed, which could be due to the FSHR saturation and downregulation.

Our study demonstrates that mouse preantral follicles from the strain employed in this study require a higher concentration of FSH to achieve larger diameters and higher antral formation rates. In contrast, the study by Xu et al. [10] showed that mouse preantral follicles only required 10 mIU/mL of FSH to reach the average diameter achieved by our high-concentration (100 mIU/mL FSH) culture. The variation could potentially be attributed to the isoform composition of FSH. The heterogeneity of FSH is primarily caused by variations in glycosylation [38], which leads to differences in isoelectric points. It has been shown that the less acidic FSH isoforms result in faster follicle growth in mice in vitro as compared to more acidic isoforms [37,38]. The FSH preparations used in the studies of mouse follicle growth have all been of human origin and it is well known that the FSH isoform distribution in human FSH preparations differ. Thus, the specific effects of the FSH used in the different studies may reflect their FSH isoform distribution.

The expression of AMH begins to increase in the primary follicles and reaches its highest level in the small antral follicles [39], which is consistent with the AMH trend in our study. The levels of AMH on days 6 and 8 were higher than that of the other two time points, though there was no significant difference between different groups. This could be due to the small number of samples examined, as the samples were from a randomly selected independent experiment conducted in the early stages of the alginate re-encapsulation technique, resulting in a lower survival rate in the 0.5–0.25% group, which is also the shortage of our study.

In conclusion, the present data provide optimised conditions for advancing mouse preantral follicle growth. Reducing hydrogel stiffness halfway through the culture, the supplementation of E_2_ in physiological concentration, and keeping the concentration of FSH at 100 mIU/mL improved the 3D culture of murine preantral follicles. Further studies should be performed to evaluate in detail the mechanisms underlying mice preantral follicle growth in vitro in connection with hydrogel stiffness, FSH, and E_2_.

## 4. Materials and Methods

### 4.1. Experiment Design

The experimental outline is shown in Figure 4. Experiment I: Follicles were randomly divided into 6 groups and cultured for 12 days [3,10] with or without E_2_ supplementation: encapsulated in 0.25% or 0.5% alginate beads for the entire culture period [10,11]; or reducing the alginate concentration (from 0.5% to 0.25%) on day 6 of culture. The medium was supplemented with 10 mIU/mL FSH (i.e., low concentration) [10]. Oestradiol was used at a concentration of 1 µM based on preliminary data where 10 nM E2 and 1 µM E2 were tested and found only an effect on follicle diameter from the high concentration_._ Experiment II: Selected follicles were encapsulated in 0.5% alginate and changed to 0.25% alginate on day 6, while E_2_ at a concentration of 1 µM and FSH at 100 mIU/mL [36] were kept constant and added to all media throughout the culture.

### 4.2. Animals 

Prepubertal female Naval Medical Research Institute (NMRI) mice (12–17 days old) were used. Animals were fed food and water ad libitum, and housed under a controlled 12 h light/12 h dark environment at 20–22 °C. The mice were sacrificed by cervical dislocation. 

### 4.3. Follicle Isolation, Encapsulation, Re-Encapsulation and Culture

Mouse ovaries were dissected free from fat and adherent tissues and cut into several pieces. Follicles were mechanically isolated from the ovaries under a stereomicroscope (Leica MZ12) using 23-gauge needles attached to 1 mL syringes. Isolation was performed in prewarmed holding medium McCoy’s 5α plus 25 mM HEPES (Gibco), 5% FBS (Gibco), and 1% penicillin/streptomycin (100×, Gibco). Following follicle isolation, follicles with diameters of 100–130 µm, an intact basal membrane, visible theca cells, and a distinct oocyte were chosen for subsequent encapsulation and culture (Figure 1A). 

Each individual follicle was pipetted into a 3 µL droplet of alginate (0.25% or 0.5%) before being placed in the cross-linking solution (50 mM CaCl_2_ and 140 mM NaCl) for 2 min. After cross-linking, alginate beads were washed in culture media: alpha minimal essential medium (αMEM) (Gibco) supplemented with 1% insulin–transferrin–selenium (100×, Gibco), 1% penicillin/streptomycin (100×, Gibco), 5% FBS, 10 mIU/mL or 100 mIU/mL human rFSH (Rekovelle, Ferring, Copenhagen, Denmark), 10 mIU/mL recombinant LH (Luveris, Serono, Germany), and 0.01% ethanol. 

Alginate beads containing single follicles were individually transferred to 96-well plates with 100 µL of culture medium containing 1 µM E_2_ (Sigma-Aldrich) in 0.01% ethanol. During the culture period, follicles were incubated at 37 °C, 5% CO_2_, and 100% humidity. Every other day, half of the culture media (i.e., 50 µL) were exchanged and stored at −80 °C for AMH measurement.

On day 6 of culture, some follicles were released from the 0.5% alginate beads by incubation in 100 μL alginate lyase solution (αMEM medium supplemented with 10 IU/mL alginate lyase, A1603, Sigma-Aldrich) for 30 min. These follicles were then re-encapsulated into 0.25% (*w*/*v*) alginate beads and cultured for 6 additional days.

Throughout isolation, encapsulation, and re-encapsulation, follicles were maintained at 37 °C. 

### 4.4. Alginate Hydrogel Preparation

Sodium alginate was obtained from Sigma-Aldrich (W201502) and prepared as previously described [40]. Activated charcoal was used to clean the alginate and to remove organic contaminants after it had been dissolved in deionised water at a concentration of 10 mg/mL. Subsequently, the alginate solution was lyophilised and sterilised. Alginate aliquots were reconstituted with sterile DPBS without Ca^2+^ (Gibco) to a concentration of 0.5 or 0.25% (*w*/*v*). 

### 4.5. Follicle Measurement

Images of each follicle were captured every second day using an inverted microscope (Zeiss Axiovert 135). Diameters of follicles were measured with the software DeltaPix 4K View (Denmark). The growth of follicles was determined by measuring the follicle diameter, the average of two perpendicular distances between the basal membrane, with one representing the largest diameter of the follicle. Follicles were considered non-viable when the oocyte was no longer surrounded by the granulosa cell layer, the granulosa cells were dark and fragmented, or the basement membrane was disrupted (Figure 1G,H). Follicles that escaped from alginate were excluded from the culture (Figure 1F).

### 4.6. AMH Hormone Assays

To evaluate the effect of different culture conditions, AMH was measured in the conditioned medium, as its secretion is related to follicle growth and development [41]. The commercially available enzyme-linked immunosorbent assay (ELISA) kits (AL-113, AnshLabs, Webster, TX, USA) developed for measuring mouse AMH were used in the study. The assay was performed according to the manufacturer’s instructions. Culture medium of one independent experiment was randomly selected and spent media from two consecutive days of collection of each follicle (days 2 and 4, days 6 and 8, and days 10 and 12) were pooled. Media collected from wells containing no follicles were used as assay controls. Samples were diluted to 1:5 for the test. 

### 4.7. Statistical Analysis

Six to nine independent cultures were performed for each culture condition. All statistical analyses were performed using the R software version 4.2.2 [42] with package sandwich version 3.0-2 [43,44] and multicomp version 1.4-20 [45]. Figures were produced using the package ggplot2 [46]. For Experiment I, follicle diameters and AMH measurements (logarithmic transformation) were compared between culture groups at different time points using two-way ANOVA model, followed by multiple testing using a maximum *t*-test (*min-p* method) adjusted for *p*-values and 95% confidence intervals. For Experiment II, comparison of follicle diameters between culture groups at different time points was performed by using Welch’s two-sample *t*-test. Follicle survival data and antrum formation data were analysed by chi-square analysis, with Bonferroni correction applied to *p*-values to account for multiple testing. All *p*-values less than 0.05 were considered statistically significant. 

## Figures and Tables

**Figure 1 ijms-24-12499-f001:**
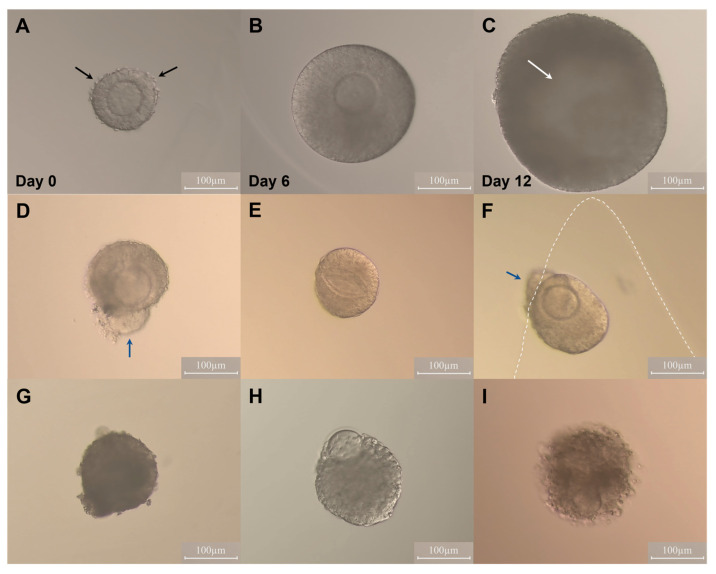
Follicle morphology during culture in a 3D alginate scaffold. (**A**–**C**) Representative images of growth and development of healthy follicles. (**A**) Two-layer secondary follicle (day 0). (**B**) Multi-layer follicle (day 6). (**C**) Antral follicle (day 12). The black arrows indicate theca cells, and the white arrow indicates the follicle antrum. (**D**–**F**) Representative images of follicles excluded from the study. (**D**) Broken basement membrane (BM). (**E**) Degenerated oocyte. (**F**) Follicle escaped from alginate and broken BM. The blue arrow indicates the location of the membrane rupture, and the dashed line marks the edge of the alginate gel. (**G**–**I**) Representative images of various non-viable follicles during culture. (**G**) Dark and dense-looking follicle. (**H**) Extruded oocyte. (**I**) Degenerated follicle. Scale bars = 100 μm.

**Figure 2 ijms-24-12499-f002:**
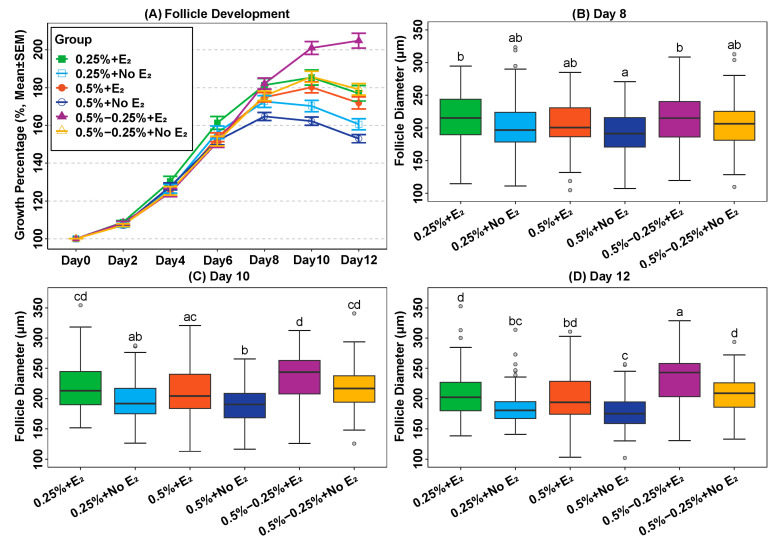
Follicle growth during culture in a 3D alginate scaffold for 12 days. (**A**) Percentage of follicle growth during the culture (**B**–**D**) Box plots display the distribution of follicle diameter in different groups on days 8, 10, and 12, respectively. The box plots show the first quartile (25th percentile), median (50th percentile), third quartile (75th percentile), maximum values, and outliers. Different letters between different groups in the boxplots indicate statistical significance. E_2_, oestradiol; SEM, standard error of mean.

**Figure 3 ijms-24-12499-f003:**
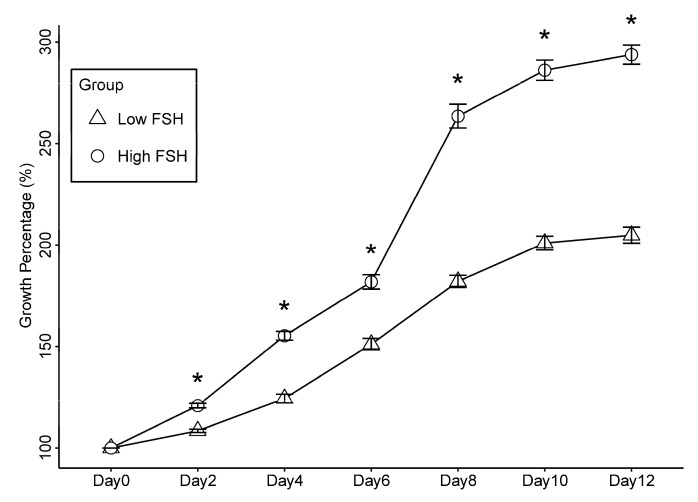
Growth of follicles cultured with 10 mIU/mL or 100 mIU/mL follicle-stimulating hormone concentrations. All follicles were embedded in 0.5% alginate gel and re-encapsulated in 0.25% alginate halfway through the culture on day 6, while oestradiol was present in a concentration of 1 µM. * Indicates statistically significant differences between two groups (*p* < 0.05).

**Figure 4 ijms-24-12499-f004:**
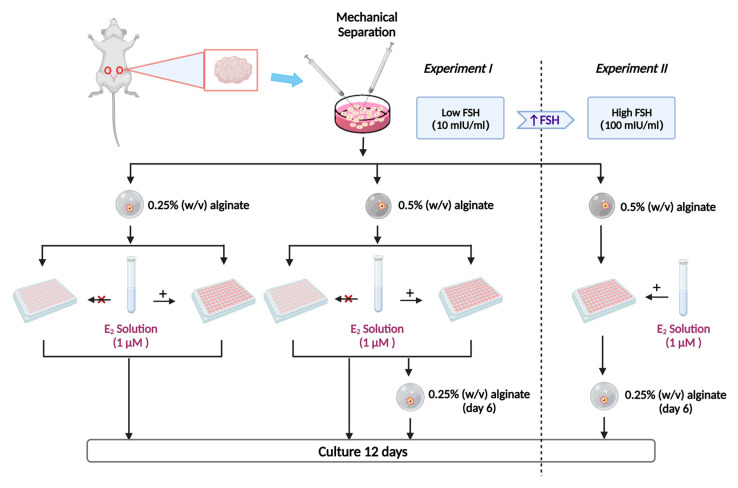
Study design. Experiment I: Follicles were randomly divided into 6 groups and cultured for 12 days. The concentration of follicle-stimulating hormone (FSH) was 10 mIU/mL and the concentration of oestradiol (E_2_) used was 1 µM: (I) encapsulated in 0.25% (*w*/*v*) alginate in a medium with E_2_ or (II); in E_2_-free medium encapsulated in 0.5% (*w*/*v*) alginate in an E_2_-containing medium (III) or in E_2_-free medium (IV); and firstly encapsulated in 0.5% (*w*/*v*) alginate for six days and re-encapsulated in 0.25% (*w*/*v*) alginate for additional six days in medium with (V) or in E_2_-free medium (VI) 1 µM. Experiment II: Follicles were encapsulated in 0.5% (*w*/*v*) alginate and re-encapsulated in 0.25% (*w*/*v*) halfway through the culture. The culture medium was supplemented with E_2_ and a high concentration of FSH (100 mIU/mL). “+” indicates E_2_ addition to the medium, while “×” indicates its absence. The figure was created with BioRender.com.

**Table 1 ijms-24-12499-t001:** Assessment parameters of follicle growth cultured in different groups in vitro.

Groups	N	Follicle Diameter (μm, mean (SD))				Survival (%)	Antrum Formation (%)
		Day 0	Day 8	Day 10	Day 12		
0.25% alginate + E_2_	85	118 (9)	214 ^b^ (39)	218 ^c,d^ (40)	208 ^d^ (40)	84% ^a^	13% ^a,b,c^
0.25% alginate + no E_2_	88	117 (8)	202 ^a,b^ (39)	198 ^a,b^ (33)	187 ^b,c^ (32)	78% ^a^	1% ^d^
0.5% alginate + E_2_	123	117 (8)	205 ^a,b^ (34)	210 ^a,c^ (38)	202 ^b,d^ (40)	72% ^a,b^	4% ^c,d^
0.5% alginate + no E_2_	125	118 (8)	193 ^a^ (30)	190 ^b^ (29)	180 ^c^ (29)	71% ^a,b^	0% ^d^
0.5–0.25% alginate + E_2_	103	117 (8)	213 ^b^ (41)	232 ^d^ (41)	233 ^a^ (42)	56% ^b^	20% ^b^
0.5–0.25% alginate + no E_2_	109	117 (8)	206 ^a,b^ (36)	215 ^c,d^ (36)	206 ^d^ (34)	54% ^b^	3% ^a,c,d^

Different superscripts within each column indicate statistically significant differences (*p* < 0.05). N, number of secondary follicles at the start of the culture; SD, standard deviation; E_2_, oestradiol.

**Table 2 ijms-24-12499-t002:** Comparisons of the average follicular diameters between different alginate concentration groups and between oestradiol and non-oestradiol groups at different time points.

	Comparison Groups	Est. Diff (µm)	95% CI	*p*-Value
Day 0	0.5–0.25% vs. 0.25%	−0.6	(−2.6; 1.4)	0.75
	0.50% vs. 0.25%	0	(−2.0; 2.0)	1.00
	0.50% vs. 0.5–0.25%	0.6	(−1.2; 2.4)	0.69
	0.5–0.25% vs. 0.25%	−0.7	(−4.4; 2.9)	0.89
Day 2	0.50% vs. 0.25%	0	(−4.2; 2.4)	0.8
	0.50% vs. 0.5–0.25%	−0.2	(−3.2; 2.9)	0.99
	0.5–0.25% vs. 0.25%	−4.8	(−13.1; 3.6)	0.37
Day 4	0.50% vs. 0.25%	−2.8	(−10.5; 4.8)	0.66
	0.50% vs. 0.5–0.25%	2	(−5.1; 9.0)	0.79
	0.5–0.25% vs. 0.25%	−9.5	(−19.8; 0.7)	0.07
Day 6	0.50% vs. 0.25%	−7.4	(−16.9; 2.2)	0.17
	0.50% vs. 0.5–0.25%	2.2	(−6.3; 10.6)	0.82
	0.5–0.25% vs. 0.25%	1.5	(−8.1; 11.0)	0.93
Day 8	0.50% vs. 0.25%	−8.8	(−17.3; −0.3)	**0.04 ***
	0.50% vs. 0.5–0.25%	−10.3	(−18.3; −2.2)	**0.01 ***
	0.5–0.25% vs. 0.25%	15.8	(5.9; 25.8)	**<0.001 ***
Day 10	0.50% vs. 0.25%	−7.7	(−16.4; 0.9)	0.09
	0.50% vs. 0.5–0.25%	−23.6	(−32.6; −14.5)	**<0.001 ***
	0.5–0.25% vs. 0.25%	22	(11.1; 33.0)	**<0.001 ***
Day 12	0.50% vs. 0.25%	−6.6	(−16.1; 2.9)	0.23
	0.50% vs. 0.5–0.25%	−28.6	(−39.0; −18.3)	**<0.001 ***
Day 0	E_2_ vs. without E_2_	0.2	(−1.1; 1.5)	0.76
Day 2		1.3	(−0.9; 3.5)	0.26
Day 4		−0.6	(−5.7; 4.5)	0.81
Day 6		2.4	(−3.8; 8.6)	0.45
Day 8		10	(4.2; 15.8)	**<0.001 ***
Day 10		19.1	(13.0; 25.3)	**<0.001 ***
Day 12		22.9	(16.1; 29.8)	**<0.001 ***

In the 0.5–0.25%: alginate group follicles were culture with 0.5% alginate during the first half of the culture (day 0 to day 6) and with 0.25% alginate during the second half of the culture (day 6 to day 12). The 0.25% and 0.5% alginate groups were encapsulated with 0.25% and 0.5% alginate during the entire culture period, respectively. * indicate statistically significant differences (*p* < 0.05) between two comparison groups. Est. Diff, the estimated difference in diameters between two comparison groups in a two-way ANOVA; 95% CI, 95% confidence interval; vs., versus; E_2_, oestradiol.

**Table 3 ijms-24-12499-t003:** AMH concentration in the condition medium between different groups at different time points.

AMH Concentration (ng/mL)	0.25% + E_2_	0.25% + No E_2_	0.5% + E_2_	0.5% + No E_2_	0.5−0.25%+ E_2_	0.5–0.25% + No E_2_
	(N = 14)	(N = 16)	(N = 10)	(N = 15)	(N = 9)	(N = 3)
Day 2 + Day 4						
Mean (SD)	3.1 (0.3)	0.9 (0.5)	0.6 (0.5)	1.2 (1.1)	3.0 (3.2)	1.7 (1.3)
Median (Min, Max)	3.3 (2.8, 3.4)	0.9 (0.1, 1.6)	0.5 (0.1, 1.3)	1.1 (0.1, 2.3)	1.8 (0.7, 9.5)	1.7 (0.8, 2.6)
Invalid tests	11 (78.6%)	10 (62.5%)	6 (60.0%)	12 (80.0%)	1 (11.1%)	1 (33.3%)
Day 6 + Day 8						
Mean (SD)	3.5 (4.0)	1.2 (1.7)	3.6 (2.9)	1.7 (1.1)	4.3 (2.7)	5.9 (1.9)
Median (Min, Max)	2.3 (0.1, 12)	0.50 (0.0, 4.2)	2.9 (1.5, 11)	1.3 (0.3, 3.9)	3.8 (0.4, 8.8)	6.9 (3.8, 7.1)
Invalid tests	6 (42.9%)	11 (68.8%)	0 (0%)	0 (0%)	0 (0%)	0 (0%)
Day 10 + Day 12						
Mean (SD)	2.7 (1.9)	1.5 (0.6)	1.1 (0.5)	1.0 (0.6)	1.5 (1.5)	2.0 (1.3)
Median (Min, Max)	2.6 (0.1, 5.5)	1.4 (0.5, 2.5)	1.1 (0.6, 1.7)	0.9 (0.1, 2.5)	1.0 (0.5, 4.4)	2.0 (1.1, 3.0)
Invalid tests	0 (0%)	0 (0%)	6 (60.0%)	2 (13.3%)	3 (33.3%)	1 (33.3%)

AMH, anti-Müllerian hormone; E_2_, oestradiol, N, number of samples; SD, standard deviation; Min, minimum value; Max, maximum value.

**Table 4 ijms-24-12499-t004:** Comparisons of AMH concentrations in the spent medium between different alginate concentration groups and groups with and without oestradiol at different time points.

	Comparison Groups	Est. Diff	95% CI	*p*-Value
Day 2 + Day 4	0.50% vs. 0.25%	−0.9	(−2.3; 0.4)	0.23
	0.5–0.25% vs. 0.25%	0.2	(−1.1; 1.6)	0.92
	0.5–0.25% vs. 0.50%	1.1	(−0.2; 2.5)	0.11
	0.50% vs. 0.25%	0.3	(−0.5; 1.1)	0.65
Day 6 + Day 8	0.5–0.25% vs. 0.25%	0.8	(−0.1; 1.7)	0.11
	0.5–0.25% vs. 0.50%	0.5	(−0.3; 1.3)	0.31
	0.50% vs. 0.25%	−0.6	(−1.1; −0.1)	**0.016 ***
Day 10 + Day 12	0.5–0.25% vs. 0.25%	−0.4	(−1.3; 0.5)	0.53
	0.5–0.25% vs. 0.50%	0.2	(−0.7; 1.1)	0.82
Day 2 + Day 4	NO E_2_ vs. E_2_	−0.6	(−1.5; 0.4)	0.24
Day 6 + Day 8		−0.5	(−1.1; 0.01)	0.054
Day 10 + Day 12		−0.3	(−0.7; 0.1)	0.18

* indicate statistically significant differences (*p* < 0.05) between two comparison groups. Est. Diff, the estimated difference in diameters between two comparison groups in a two-way ANOVA; 95% CI, 95% confidence interval; vs., versus; AMH, anti-Müllerian hormone; E_2_, oestradiol.

**Table 5 ijms-24-12499-t005:** Comparison of follicle growth evaluation parameters between the 10 mIU/mL and 100 mIU/mL FSH concentration groups.

Groups	N	Survival (%)	Follicle Diameter (μm, Mean (SD))							Antrum Formation (%)
			Day 0	Day 2	Day 4	Day 6	Day 8	Day 10	Day 12	
Low FSH group (10 mIU/mL)	103	56%	117 (8)	127 ^a^ (15)	146 ^a^ (32)	178 ^a^ (41)	213 ^a^ (41)	232 ^a^ (41)	233 ^a^ (42)	20% ^a^
High FSH group (100 mIU/mL)	33	67%	118 (6)	142 ^b^ (12)	183 ^b^ (21)	215 ^b^ (31)	311 ^b^ (48)	337 ^b^ (40)	343 ^b^ (34)	82% ^b^

Different letters show significant differences in the same column. N, number of secondary follicles at the start of the culture; SD, standard deviation; FSH, follicle-stimulating hormone.

## Data Availability

The data presented in this study are available upon request from the corresponding author.

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
