# Peer review of "Reducing 3D Hydrogel Stiffness, Addition of Oestradiol in a Physiological Concentration and Increasing FSH Concentration Improve In Vitro Growth of Murine Preantral Follicles"

_ijms, 2023, doi:10.3390/ijms241512499_

Round 1

Reviewer 1 Report

Comments and Suggestions for Authors

Reviewer 2 Report

Comments and Suggestions for Authors

Comments to the authors

The study by Zheng et al. explored the effect of hydrogel stiffness, E2, and FSH levels on preantral follicles growth and survival in mice. Overall, the study sounds sense, but the experimental design was too simple, I would suggest adding mechanism analysis if it is possible to enrich this study with more evidence.

Moreover, in discussion section, the author should give possible underlying mechanisms to explain their results. Currently, there is comparison and contrast between their findings and previous studies; however, there is no explanation why their results is different than other researchers.

Author Response

The study by Zheng et al. explored the effect of hydrogel stiffness, E2, and FSH levels on preantral follicles growth and survival in mice. Overall, the study sounds sense, but the experimental design was too simple, I would suggest adding mechanism analysis if it is possible to enrich this study with more evidence.

Moreover, in discussion section, the author should give possible underlying mechanisms to explain their results. Currently, there is comparison and contrast between their findings and previous studies; however, there is no explanation why their results is different than other researchers.

We have now improved the discussion section by including more mechanistic analyses and agree that this has improved the manuscript. 

Reviewer 3 Report

Comments and Suggestions for Authors

This study identifies three parameters that optimize 3D mouse preantral follicle cultures: the addition of E2 in a concentration of 1 µM to the culture medium, the renewal of the alginate scaffold reducing the concentration to half midway through the culture period, and the increasing of FSH concentration from 10 to 100 mIU/mL throughout the culture period.

The manuscript is clear and well written. The results are described in details and figures are well designed.

I would point to the following issues.

1. the authors should provide evidence of the ethical approval for designing a study implying animals

2. the authors should clarify how they select the timings and concentrations used in the study. Particularly regarding, E2, FSH and alginate. didi they perform preliminary results?

Comments on the Quality of English Language

 Minor editing of English language required

Reviewer 4 Report

Comments and Suggestions for Authors

Dear Authors,

The manuscript entitled "Reducing 3D hydrogel stiffness, addition of oestradiol in a physiological concentration and increasing FSH concentration improve in vitro growth of murine preantral follicles." is a very well conducted study.

I do not have any specific comments to perform for this study.

The authors could include in their discussion section the following publication doi:10.3390/medicines10030019 

Well done!!

  •  

Author Response

Response: Thank you for your review. This reference has now been included in the manuscript (Line 272).

Round 2

Reviewer 1 Report

Comments and Suggestions for Authors

Author Response

Comment for the authors (Revised version)
Thank you for addressing all comments in the revised version. Almost all the comments are sufficiently improved. However, some comments still require better clarification.
• The AMH experiment revealed a notably low number of follicles, particularly in the 0.5-0.25+No E2 group (N=3) compared to the other treatment groups.
Response: We randomly selected one independent experiment measuring AMH in conditioned media, which we believe is representative. This experiment was conducted when the technique of re-encapsulation with alginate was at an early stage, resulting in a lower survival rate in the 0.5%-0.25% groups. Indeed, this limitation represents an area for improvement in our study and we state it in our discussion section in lines 273-274.
Comment after revision: The statement in lines 273-274, "Our study demonstrates that the strain of mouse preantral follicles used in this experiment requires a higher concentration of FSH to achieve larger diameters and higher antral formation rates," does not provide an explanation for the reason behind using a low number of follicles in the 0.5-0.25+No E2 group.
Response: Sorry for the incorrect line numbers mentioned in our previous response. After submitting our initial response to your review comments, we received comments from a new reviewer, which led to further revisions to the manuscript. Unfortunately, we were unable to withdraw our initial response, which resulted in the misplaced line numbers. Regarding the limitation of the small sample size, we now provide a discussion addressing this concern in lines 292-296 of the revised manuscript.

• Figure 4 presents a confusing diagram as it incorrectly displays only one treatment group for experiment 2.
Response: Figure 4 displays the experiment correctly. However, the description of the experiment has not been accurate. In Experiment II, we improved the experimental conditions by increasing the FSH concentration to 100 mIU/ ml and cultured 33 follicles. We revised the description in the Methods and Materials.
Comment after revision: Based on my understanding, the purpose of experiment 2 was to investigate whether increasing FSH concentration (from 10 to 100 mIU/ml) would promote follicle development. However, the revised version in lines 316-319 only mentions one group in this experiment (0.5-0.25 + E2 + high FSH) and omits the "10" mentioned in the previous version. Although the author mentioned that the figure was displayed correctly, I could only identify one treatment group in experiment 2. I recommend the author to revise this section to provide better clarification. 
Response: In Experiment II, we conducted a single experiment, in which follicles were cultured under the conditions of re-encapsulation with alginate, addition of E2 and a high concentration of FSH. The results in this experiment were compared with those of the previous experiment I in which follicles were cultured under conditions of re-encapsulation with alginate, addition of E2 and a low concentration of FSH. We revised the description in M&M section lines 312-316 and Results section lines 154-157.

€ The authors should provide an explanation for measuring AMH in the manuscript.
Response: The AMH measurement was described in lines 346-353.
Comment after revision: The author should provide an explanation for their decision to evaluate the AMH level in this study. This will help readers better understand the significance of measuring AMH and its importance in the context of this research.
Response:  Sorry for the incorrect line numbers. We stated the reason for measuring AMH in lines 375-378, and the reference was also added.

Reviewer 3 Report

Comments and Suggestions for Authors

The maniscript has been revised and improved according to my suggeations. It can be considered for publication in the current form

Comments on the Quality of English Language

Minor editing of English language required

Author Response

The maniscript has been revised and improved according to my suggeations. It can be considered for publication in the current form.

Response: Thank you for your valuable feedback.